# Genome-Wide Characterization of Soybean 1-Aminocyclopropane-1-carboxylic Acid Synthase Genes Demonstrates the Importance of *GmACS15* in the Salt Stress Responses

**DOI:** 10.3390/ijms26062526

**Published:** 2025-03-12

**Authors:** Peng Cheng, Yi-Cheng Yu, Si-Hui Wang, Jun Yang, Run-Nan Zhou, Xin-Ling Zhang, Chun-Yan Liu, Zhan-Guo Zhang, Ming-Liang Yang, Qing-Shan Chen, Xiao-Xia Wu, Ying Zhao

**Affiliations:** 1National Key Laboratory of Smart Farm Technology and System, Key Laboratory of Soybean Biology in Chinese Education Ministry, Northeast Agricultural University, Harbin 150000, China; naturechengpeng@163.com (P.C.); wshui206@163.com (S.-H.W.); 18591620129@163.com (J.Y.); 19853566679@163.com (X.-L.Z.); cyliucn@126.com (C.-Y.L.); zhanguo7907@126.com (Z.-G.Z.); mlyang@neau.edu.cn (M.-L.Y.); qshchen@126.com (Q.-S.C.); 2School of Life Sciences and State Key Laboratory of Agrobiotechnology, The Chinese University of Hong Kong, Sha Tin, New Territories, Hong Kong, China; yuyicheng007@gmail.com; 3Heilongjiang Academy of Agricultural Sciences, Nangang District, Harbin 150000, China; runnanzhou@126.com; 4Heilongjiang Green Food Science Research Institute, Harbin 150000, China

**Keywords:** *Glycine max* L., 1-aminocyclopropane-1-carboxylic acid synthase, salt stress

## Abstract

ACS (1-aminocyclopropane-1-carboxylic acid synthase) is a member of the aminotransferase superfamily and a pyridoxal phosphate-dependent enzyme. ACS is also a rate-limiting enzyme for the biosynthesis of ethylene and has been linked with plant development, growth, and stress responses. However, information on *ACS* genes in the soybean genome is limited. In this study, we identified *ACS* genes in soybean through phylogenetic trees and conserved motifs and analyzed their cis-acting elements, subcellular localization, and expression patterns. Twenty-two members of the *ACS* family were identified in soybean, and they were divided into four subfamilies based on phylogenetic relationships. Moreover, the results of Arabidopsis mesophyll protoplasts showed that *GmACS1*, *GmACS8*, and *GmACS15* were all localized in the nucleus and cell membrane. Cis-regulatory elements and qRT-PCR analyses indicated markedly increased levels of *GmACS* transcripts under hormone treatments and abiotic stress conditions (drought, alkalinity, and salt). In addition, under different abiotic stresses, the potential functional variations across the *GmACS* isoforms were mirrored in their differential expression. The analysis of transcriptional response to salinity indicated that salt stress might primarily be mediated by the *GmACS15* gene. *GmACS15* was also found to reduce salt-induced oxidative damage by modulating the ROS-scavenging system, cellular redox homeostasis, and maintaining intracellular Na^+^/K^+^ balance. The results of this investigation revealed the involvement of the *ACS* gene family in soybean stress-response pathways, including the identification of a potential target for enhancing salt tolerance in soybean.

## 1. Introduction

Soybean, as one of the most important crops globally, is crucial for providing humans with substantial plant proteins and oils. According to recent data, soybeans are primarily consumed in three main ways: direct human consumption (20%), animal feed (76%), and industrial uses (4%) [1]. However, the escalating impact of biotic and abiotic stresses due to global climate change poses significant challenges to soybean yield and quality. It has been estimated that drought stress alone can reduce soybean yields by up to 50% in affected regions [2]. Therefore, a detailed analysis of these stress factors and their effects on soybean physiology is essential. Exploring stress-related genes in soybeans can effectively enhance their stress resistance mechanisms and alleviate the adverse effects of unfavorable conditions on soybean growth. We aim to focus on the identification and functional analysis of stress-responsive genes and provide a foundation for breeding more resilient soybean varieties.

Ethylene is a crucial hormone found in plants, significant for different physiological mechanisms, such as leaf senescence, seed germination, fruit ripening, and the plant’s response to environmental stressors [3]. Ethylene biosynthesis modulation is essential for plant survival during abiotic stresses, including extreme temperatures, pathogen attacks, salinity, and drought [4,5]. Ethylene biosynthesis comprises two primary steps. In the first step, 1-aminocyclopropane-1-carboxylic acid synthase (ACS) catalyzes the ethylene precursor 1-aminocyclopropane-1-carboxylic acid (ACC) production from the substrate S-adenosyl methionine (SAM). Then, in the 2nd step, the ACC oxygenate (ACO) oxidizes ACC to ethylene, CO_2_, and cyanide [6,7]. The rate-limiting step in this process is the conversion of SAM to ACC catalyzed by ACS. Therefore, for ethylene synthesis in plants, ACS is the primary rate-limiting enzyme. A recent study indicated that in plants, biotic or abiotic stress can enhance *ACS* gene levels, thus altering the ethylene content, activating the ethylene-mediated signal pathway, and modulating downstream gene levels to improve stress resistance. Consequently, numerous *ACS* genes have emerged as a focal point of research. Nevertheless, the functional characterization of *ACS* genes in soybean in response to abiotic stresses, such as salinity, remains equivocal [8].

*ACS* is an aminotransferase superfamily enzyme, which is pyridoxal phosphate-dependent and has a multigene subfamily in higher plants. Although they are a divergent multigene family, they comprise a similar molecular size primary structure (441 to 496 amino acids). In addition, their PLP-binding site is in Region 5, which is crucial. Moreover, their N-terminal end is a markedly conserved region comprising serine and leucine residues, whereas the C-terminus is a hypervariable site of 18 to 85 residues and is considered the core region for subfamily classification. Currently, with the advancements in genome analysis technology, many *ACS* gene family members have been determined in various plant species. For example, 12 *ACS* genes were found in Arabidopsis [9], 9 in tomato [10], 6 in rice [11], 12 in wheat [8], 10 in grapes, and 11 in poplars [11]. However, the *ACS* family members in soybeans have not been identified or analyzed.

The literature suggests that *ACSs* are crucial for plant development, growth, and responses to abiotic and biotic stresses [12]. Several research studies have shown the association of *AtACS2* in the lateral root development of Arabidopsis and *AtACS2* overexpression notably reduced the number of lateral roots [13,14,15]. Furthermore, in maize, under normal growth conditions, silencing *ZmACS6* expression delays leaf senescence, thereby enhancing its resistance to drought [16]. Moreover, it has been observed that hypoxia stress treatment substantially enhanced *OsACS5* expression in rice. In watermelon, pumpkin, cucumber, and melon, the homologous genes *CsACS2*, *CpACS27*, *CitACS4*, and *CmACS7* influenced sex differentiation by modulating flower growth [17,18].

Recent studies have highlighted the importance of *ACS* genes in soybean, particularly in response to biotic and abiotic stresses. For instance, Tucker et al. (2010) reported that the expression of *ACS* genes and ACC concentration in soybean roots and root tips are significantly influenced by environmental factors and biotic interactions [19]. While Tucker focused on the role of *ACS* in soybean roots under nematode infection, our study employs genome-wide analysis to systematically investigate the expression patterns of the soybean *ACS* gene family under multiple stress conditions. Additionally, we explore the structural characteristics, evolutionary relationships, and functional divergence of this gene family under various stress conditions. This not only fills the gap in the systematic study of the soybean *ACS* gene family but also provides new perspectives and theoretical bases for a deeper understanding of their roles in plant adaptation to complex environmental changes.

## 2. Results

### 2.1. Identification of ACS Family Genes and Sequence Analysis in Soybean

To determine the *GmACS* gene family members, the amino acid sequence data of *ACS* in rice, wheat, and Arabidopsis in the Phytozome v12.1 database [8] were compared. In total, 22 *GmACS*s (termed *GmACS1–22*) were identified and characterized in the soybean genome. Appendix A lists detailed information including molecular weight (MW), gene ID, protein length (aa), gene location, and theoretical isoelectric point (pI). The gene length of *GmACS1–22* ranged between 372 bp (*GmACS4*) and 1590 bp (*GmACS14*), and whole-length CDS encoded 22 putative proteins comprising 124 to 530 amino acids. The molecular mass and protein PI of 22 *GmACS*s ranged from 5.63/13.85 to 9.08/58.58 kDa (Appendix A).

To investigate the evolutionary relationship of *ACSs* in *Oryza sativa*, *Arabidopsis thaliana*, *Glycine max*, *Medicago truncatula,* and *Triticum aestivum*, a phylogenetic tree for 61 *ACSs* of these five species was constructed (9 in *Arabidopsis thaliana*, 22 in *Glycine max*, 6 in *Oryza sativa*, 12 in *Medicago truncatula,* and 12 in *Triticum aestivum*). The *ACSs* were compared with their homologs in *Arabidopsis thaliana*, *Medicago truncatula*, *Triticum aestivum* L., *Glycine max*, and *Oryza sativa,* which revealed that plant *ACSs* were segregated into four specific groups according to their phylogenic relationships (Figure 1A). Based on the branches, all 22 *ACS* genes were categorized into 4 subfamilies. According to Arabidopsis and rice classification, *GmACS*s had the same branches in rice and Arabidopsis as subfamilies I–III. Subfamily I had the largest number of soybean members, with eight genes, whereas subfamilies II and III contained six and four genes, respectively. Furthermore, based on the classification of wheat *ACS* genes that lack catalytic activity, *GmACS17*/*3*/*14*/*20*, *MtACS4*, *OsACS6*, and *TaACS8*/*11*/*12* were categorized into the same branches and collectively assigned to subfamily IV.

This study also assessed and compared the sequences of 22 GmACS proteins; the genomic sequence length was between 518 bp (*GmACS4*) and 5645 bp (*GmACS9*) (Figure 1B), indicating that all members comprised the typical Aminotran_1_2 domains. Subfamilies III and IV genes contained three and four exons, respectively. In subfamily I, one gene (*GmACS4*) contained two exons and the remaining *GmACS* genes contained four exons. In subfamily II, one gene (*GmACS5*) contained two exons, one gene (*GmACS9*) contained four exons, and the remaining *GmACS* genes contained three exons. This conservation might be because of introns and exon conserved structure caused by gene recombination and replication during the evolution of this gene family.

### 2.2. Syntenic Relationship Assessment of GmACSs

To assess how *GmACS* members evolved, the *ACS* genes of *Glycine max*, *Arabidopsis thaliana*, *Medicago truncatula*, *Oryza sativa*, and *Triticum aestivum* were compared. In soybeans, 12 of 20 chromosomes contained the 22 *GmACS*s (Figure 1C). The aforementioned five species shared 123 *ACS* orthologous pairings (Figure 1C). Furthermore, 10 orthologous gene pairs were observed between *Glycine max* and *Arabidopsis thaliana,* 11 between *Triticum aestivum* and *Glycine max*, and 17 between *Glycine max* and *Medicago truncatula* (Figure 1C). In addition, the soybean genome had 49 paralogous *ACS* gene pairs (Figure 1D). These duplicate genes were often associated with subfamilies or species (Figure 1C). This observed synteny proved that various *ACS* genes had evolved prior to the divergence of soybean species (Appendix A).

### 2.3. Regulatory Elements in the GmACS Promoters

Cis-acting elements are crucial for modulating gene transcription. To evaluate the potential role and transcriptional modulation mechanism of these genes, a 2000 bp sequence upstream of each gene was selected to analyze and predict the cis-elements in the promoters. Then, the existence of cis-elements associated with *GmACS* transcriptional control was assessed. It was observed that most of the *GmACS* genes had hormone-responsive elements that were further classified into four distinct groups: light-responsive elements (G-box), environment stress-related elements [TC-rich, anaerobic inducible element (ARE), LTR, and MBS], hormone-responsive elements [TCA-element, TGA-element, hormone-responsive elements (ABRE), TGACG, and CGTCA], and development elements (O2-site) (Figure 1E). Furthermore, most *GmACS*s, except *GmACS4* and *GmACS11*, contained the cis-acting element G-box. During stress conditions such as high salt, drought, and low temperature, the G-box can interact with the corresponding transcription factors to activate downstream genes and initiate stress-resistance responses in plants. *GmACS2*, -*6*, -*9*, -*11*, -*15*, and -*18* contained the hormone-responsive element, TCA-element. TCA-element can respond to stress signals, regulate related gene expression, and help plants adapt to adverse environmental conditions. In addition, the promoter sequences of many *GmACS* genes had cis-elements associated with the environment stress-related elements, including ARE which was identified in *GmACS3*, -*4*, -*7*, -*8*, -*9–12*, -*14–17*, -*19*, -*20*, and -*22*. The cis-element ABRE was identified in all *GmACS*s except *GmACS1*, -*2*, -*4*, -*8*, -*11*, -*13*, and -*17*. Overall, these bioinformatics results suggest that *GmACS*s are involved in regulating plant development and stress responses.

### 2.4. GmACS’s Subcellular Localization

The CDS regions of three *GmACS* genes were cloned, and their accession numbers were submitted to GenBank. The coding regions of three *GmACS* genes [*GmACS1* (PQ866053), *GmACS8* (PQ866030), and *GmACS15* (PQ866067)] were fused to the GFP coding site’s N-terminus to construct pSOY1-*GmACS* to create a transient expression in Arabidopsis protoplasts. The *GmACS* genes were all located in the nucleus and on the cell membrane. The positive control (35S::GFP) was present in all cell compartments except the vacuole and chloroplast (Figure 2). These data are consistent with the predictions of online websites and observed *ACS* genes’ subcellular localization in wheat.

### 2.5. GmACS Expressions in Various Tissues During Various Abiotic Stresses and Hormone Treatments

Different *GmACS* genes are involved in different stages of soybean growth and development, which can be observed from their tissue-specific expression patterns. It was observed that *GmACS* gene transcripts were present in stems, root nodules, leaves, pods, em, roots, mm, lm, ds, and flowers. Furthermore, the expression of different genes varied markedly among the above 10 tissues during soybean growth and development. *GmACS11*/*GmACS13*/*GmACS18* showed relatively high expression levels in roots. Moreover, the transcriptional levels of *GmACS15*/*10*/*1* were very high in flowers compared to other genes, whereas the expression of *GmACS8*/*14*/*20* was very high in leaves (Figure 3).

To further analyze their possible roles in abiotic stress, the *GmACS* family’s expression patterns under salt stress (150 mmol/L sodium chloride), alkali stress (100 mmol/L sodium bicarbonate), and osmotic stress (20% polyethylene glycol or 200 mmol/L mannitol) were observed. The data revealed that under saline–alkali stresses, the expression of most *GmACS* genes was notably increased (Figure 4A). Furthermore, the expression of genes *GmACS1*, *GmACS8*, *GmACS13*, *GmACS15*, *GmACS18*, and *GmACS19* was first upregulated and then downregulated. Moreover, *GmACS15* indicated the highest expression after salt stress. After 12 h of salt treatment, the *GmACS15* expression level was substantially higher than other genes and was upregulated > 100 times. In addition, its expression was also high after 6 h of alkali treatment, indicating an expression increase of about 70 times. Compared with other conditions, the degree of change in *GmACS* transcripts following osmotic treatment (polyethylene glycol (20%) or mannitol (200 mmol/L)) was lower. Only a few genes such as *GmACS1*, *GmACS9*, *GmACS12,* and *GmACS15* showed a small increase in expression after exposure to 200 mmol/L PEG. After 200 mmol/L mannitol stress, the response degree of *GmACS*s is relatively small (Figure 4A).

Most identified *GmACS* genes have been observed to be modulated by cis-acting elements related to hormone signals (ABRE, TCA, TGA, TGACG, and CGTCA) (Figure 1E). Therefore, *GmACS* transcriptional abundance in response to SA, ABA, JA, and BR treatments was observed to assess the potential association of these enzymes in plant hormone-mediated signal transduction (Figure 4B). qRT-PCR analysis revealed that most *GmACS*s respond to all hormone treatments; however, *GmACS* genes indicated high variability in their responses to these different hormone treatments. Under SA treatment, *GmACS7* showed the most significant response. Under JA hormone stress, *GmACS10* and *GmACS19* had the most notable responses. Under ABA hormone stress, levels of *GmACS8* were substantially increased throughout the treatment period. Compared with other hormones, after Br treatment, the transcripts of *GmACS1*, *GmACS10*, *GmACS11*, *GmACS12*, *GmACS18*, and *GmACS19* markedly increased after 3 h of stress treatment and reached the peak after 6–12 h. Moreover, *GmACS15* gene expression started to enhance after 6 h of treatment and remained high after 24 h of stress, and the expression was substantially upregulated by about 60 times. These results show that *GmACS1*, *GmACS8,* and *GmACS15* had a strong response against salt stress among which *GmACS15* showed the most obvious reaction; therefore, it was selected for subsequent research.

### 2.6. GmACS15 Overexpression Enhances Salt Tolerance

This research study purified the GST-tagged *GmACS15* plasmid-encoded crude proteins in *E. coli* cells to assess *GmACS15’s* resistance activities. SDS-PAGE was carried out to identify the GmACS15-GST fusion protein’s molecular weight (Figure 5A). Furthermore, the enzyme kinetic analysis of recombinant GmACS15 protein was conducted using PLP and SAM as substrates. The kinetic parameters were depicted by Eadie–Hofstee plots. The acquired Vmax values for SAM and PLP were 0.61 and 1.39 µmol/min/mg protein, respectively, whereas the Km values were 0.37 and 0.41 mmol/L, respectively (Figure 5B,C). These results indicated that the purified GmACS15 protein had ACS enzyme activity.

To understand how *GmACS15* modulates salt stress response, composite soybean plants with non-transgenic leaves and *GmACS15* overexpressing transgenic HRs were produced. The enzyme activity and expression of *GmACS15* were elucidated in *GmACS15* overexpressing HRs (OHR1-OHR8), which revealed that *GmACS15* transcripts were 5.3–13.1-fold higher than in control HRs (CHR) (Appendix A). Moreover, the enzymatic test revealed that in OHR, the *ACS* activity was 4.5–16.8-fold higher than in CHR (Appendix A). These data suggested that the *GmACS15* gene had ACS activity and was successfully overexpressed in transgenic soybean hair roots. For subsequent analyses, two OHR lines with high *GmACS15* expression (OHR5 and OHR6) were selected. *GmACS15* overexpression indicated no significant effect on soybean growth, and there was no abnormal growth phenotype in composite plants with transgenic HRs or CHR under standard conditions (Figure 5D). On the third day of NaCl treatment, the length and dry and fresh weight of the aboveground and belowground parts of each line were measured. The *GmACS15*-OHR plant phenotypes after salt treatment were different from the CHR plants. Relative to the CHR, OHR plants had stronger osmotic/salt resistance, as evidenced by fresh root weight and enhanced root elongation (Figure 5E,F). The OHR plant’s aboveground parts had heavier fresh leaf weight; however, there was no substantial difference between CHR and OHR plant height (Figure 5G,H). Overall, these results indicated that *GmACS15* overexpression can enhance the soybean plant’s tolerance to salt.

### 2.7. GmACS15 Modulates ROS Levels Under Salinity

To further examine the ROS levels under salinity, mature leaves of CHR and *GmACS15* transgenic hair roots were dyed with DAB to identify H_2_O_2_ and NBT to access the production of superoxide radicals. Faint and sporadic staining in *GmACS15*-OHR leaves under salt indicated reduced ROS production. Evan’s blue staining indicated salt stress-induced cell death. Samples under non-stress conditions showed mild intensities of Evan’s blue (Figure 6A). Furthermore, salt stress moderately induced ROS levels in CHR leaves and slightly induced in OHR leaves. Moreover, H_2_O_2_, O_2_^−^, and MDA levels were markedly reduced in the CHRs and leaves of OHR plants relative to those of the CHR plants. This research study also evaluated the antioxidant enzyme activity and metabolites involved in REDOX exchange in the HRs of transgenic *GmACS15* in a high-salt environment. It was found that the activities of SOD, POD, and CAT in *GmACS15*-OHR plants were enhanced more prominently than those in CHR plants. Moreover, these metabolite levels did not alter between CHR and OHR during normal conditions. However, during salt stress, the OHR plant had increased GSH and reduced GSSG levels than CHR, resulting in increased GSH/GSSG ratios (Figure 6C).

Propidium iodide (PI) is a nuclear stain that can be used to study cell damage or death. Non-stress condition samples indicated mild PI stain intensities (Figure 6), whereas the root tips had a deep PI stain, which indicated that the root tips had a substantially elevated number of dead cells after salt treatment. The cell death rate was reduced in OHR plants than in the CHR plants, indicating that *GmACS15* overexpression inhibited oxidative stress-induced cellular damage (Figure 6B). During salt stress, the transmembrane flow of potassium (K^+^) and sodium (Na^+^) flow is crucial for plant adaptation. Here, a fluorescent sodium ion probe-CoroNa™·Green was employed to monitor changes in the concentration of intracellular Na^+^ and measure the Na^+^ and K^+^ content. The acquired results showed that the fluorescence intensity of CHR and OHR was weak under normal conditions and the difference was not statistically substantial. Under salt stress, CHR showed significantly stronger CoroNa-Green fluorescence than OHR. Moreover, the comparative analysis reveals that OHR roots accumulate less Na^+^ than CHR. Furthermore, during salt stress, the OHR accumulated lower Na^+^ than CHR, while K^+^ indicated no significant change, leading to lower Na^+^/K^+^ ratios (Figure 6B,C). These results suggest that *GmACS15* may enhance salt tolerance by maintaining intracellular Na^+^/ K^+^ balance.

For an in-depth assessment of how mitochondrial *GmACS15* affects redox and ROS homeostasis, the expression of various known participating genes was investigated. In the absence of salt stress, no significant variations were observed in the transcripts of all tested genes. However, the salt stress markedly induced the mRNA levels of OHR more than that of CHR (Figure 7). These findings demonstrate that *GmACS15*-OHR had elevated levels of stress-inducible responses of these genes, which potentially caused stress-resistant phenotypes of OHR plants.

## 3. Discussion

*ACS* is a naturally occurring enzyme that is the crucial and rate-limiting enzyme for the ethylene biosynthesis pathway [20]. Furthermore, it modulates signal transmission which is crucial for plant development and growth. Several studies have identified genes responsive to various abiotic and biotic stresses, such as drought stress, fungal elicitor, salt stress, wounding, freezing, and anaerobic conditions [21,22,23,24]. Most current studies on *ACS* genes are focused on plants including wheat [8], quinoa [25], grapes, and poplars [11]. Recent investigations have delved into certain facets of the *ACS* gene family in soybeans. Nevertheless, a comprehensive and in-depth comprehension of the entirety of the *ACS* gene family in soybeans has yet to be achieved, thus necessitating further exploration. Here, 22 *ACS* genes, termed *GmACS1-22*, were identified in the soybean genome (Appendix A). Furthermore, the encoded GmACS proteins had the same specific and essential protein domains (PF00155) as other typical ACSs.

The phylogenetic tree identified 61 ACS proteins in different plants (Figure 1), including *M. truncatula*, *G. max*, *A. thaliana,* and others. These data indicated that ACS proteins are broadly distributed in different higher plants. Based on their distribution in various subfamilies, it was inferred that the *ACS* family has substantially expanded throughout evolution and has been crucially associated with higher plant evolution. The comprehensive assessment of the soybean genome identified 22 putative *ACS* genes (Appendix A), which were classified into four phylogenetic branches per their phylogenetic relationships, subcellular location, and protein structures, indicating certain functional divergence in these related family members throughout *G. max*’*s* evolution. Consistent with their genetic association, the *GmACS*s were found to be substantially linked with the homologous genes in leguminous plants, such as *M. truncatula*, which suggested that they have potential functional protective properties (Figure 1). This investigation also found that *GmACS*s are collinearly related with *ACSs* from *A. thaliana*, indicating that *GmACS*s might have been produced before Arabidopsis divergence (Figure 1). Moreover, all 22 *GmACS* genes had significant homogeneity in the distribution and length of exons and introns in the same subgroup, demonstrating that *ACSs* are evolutionarily conserved. *GmACS1*, -*8*, and -*15* were identified as the three successfully expressed fusion proteins. In line with the online predicted results, *GmACS1*, *-8*, and -*15* were specifically localized in the cell membrane and nucleus. It was, therefore, suggested that these *GmACS* isoforms have distinct roles in various organelles. These results provided crucial data for novel hypotheses on the role of *ACS* members in plants.

Several recent studies have investigated new *ACS* functions in maize [16], *A*. *thaliana* [26], carnation [27], and cucumber [17], such as their involvement in post-germination growth, storage reserve accumulation, embryo development, cellular energetics, and seed maturation. Here, various *GmACS* gene expression patterns were observed in different tissues, indicating that they have functionally distinct roles in soybeans. Most *GmACS*s (40.9%) had increased expression in roots and nodules, whereas 27% had transcripts, specifically abundant in flowers and leaves (Figure 3). The physiological association of *GmACS*s with soybean growth was confirmed by their expression trend in different tissues. This investigation also analyzed the transcriptional responses of *GmACS*s to different stresses such as alkali, hormonal stimuli, osmotic stress, and salt stress. The results revealed that these stresses, especially salt stress, notably induced the expression levels of *GmACS*s (Figure 4). It has been observed that *GhACSs* [22], *CqACSs* [25], and *TaACSs* [8] transcripts are substantially elevated during stressful environments including dehydration, cold, and salt. In addition, the PEG6000 and mannitol-induced osmotic stressors indicated various effects on *GmACS*s transcription and expression, potentially because PEG inhibits the root oxygen supply. Here, it was found that the *GmACS* gene’s promoter regions comprised numerous elements related to abiotic and biotic stress responses, such as stress-response elements (ARE, TC-rich, LTR, and MBS recognition sites) and hormone-responsive elements (ABRE, TGA-element, and TCA-element), indicating that these genes are under intricate regulatory control. The differential expression of *GmACS* genes under various stress conditions is likely attributed to the combined action of these cis-elements and their interacting transcription factors (Figure 1). Future research could focus on identifying the specific transcription factors that bind to these elements and elucidating the molecular mechanisms underlying their regulatory functions. Overall, the data strongly suggest that *GmACS* genes are potentially associated with the responses of plants to different abiotic and biotic stresses. The rapid upregulation of *GmACS* genes, especially *GmACS15* under salt stress, implies the involvement of signaling pathways that rapidly transduce stress signals into transcriptional responses.

*GmACS15* responded more strongly and rapidly to salt stress relative to other genes (Figure 4) and reached the highest transcriptional level (about 120-fold) within 12 h of salt treatment, performing a significant part in salinity response. Therefore, *GmACS15* was further analyzed to determine its underlying molecular pathway in mediating salinity adaptation. Subsequently, the assessment of enzymatic functions of *E. coli*-expressing recombinant GmACS15 proteins demonstrated that GmACS15 had an affinity for substrates (Km^SAM^ of 0.61 mmol/L) (Figure 5) in line with previously indicated kinetic parameters in ACSs [28]. Fresh root weight and root length were markedly increased in *GmACS15* overexpressing transgenic soybean hair roots, indicating increased salt tolerance of the plant (Figure 5).

The literature suggests that the stimulation of the antioxidant system during abiotic stress eliminates increased ROS [29]. Abiotic stress can induce a REDOX imbalance in plants, resulting in oxidative stress and lipid peroxidation in cells. Therefore, it is crucial to maintain REDOX balance in plants for stress resistance [30]. Salt stress conditions increase SOD, POD, and CAT activities. However, the activity of *GmACS15*-OE plants increased more significantly than that of WT plants. Furthermore, NBT, DAB, and Evan’s blue staining were carried out on the transgenic plant leaves. The staining revealed the H_2_O_2_ and O_2_^−^ contents, as well as the cell damage of plants. *GmACS15*-OHR staining was lighter and the H_2_O_2_ and O_2_^−^ content was lower, indicating that *GmACS15* is involved in the ROS detoxification process during salt stress (Figure 6A). To further verify the extent of plant damage, PI staining was performed on root tips after stress, which demonstrated that *GmACS15* overexpressing plants had the lightest PI staining and fewest staining sites, suggesting that *GmACS15* overexpression can reduce the degree of cell death and cell membrane damage in plants to resist salt stress (Figure 6B). Under salt–alkali stress, high Na^+^ concentration in the soil causes an imbalance of ion balance in plant cells, damages cell membrane structure, reduces metabolic activity, and leads to ion toxicity [31]. Moreover, the CoroNa™ Green Na^+^ ion indicator was employed to stain WT and overexpressed strains after stress induction and measured Na^+^ and K^+^ content. The data revealed that during salt stress, overexpressed plants had lighter color and lower Na^+^ and K^+^ levels than WT strains, suggesting that *GmACS15* overexpression can increase Na^+^ ions efflux from plant roots and maintain Na^+^ and K^+^ balance to enhance plant salt tolerance (Figure 6B). *GmACS* genes might be part of a larger gene regulatory network (GRN) that integrates multiple stress signals and coordinates cellular responses. Identifying the upstream regulators and downstream targets of *GmACS* genes within this network will be essential for understanding their role in stress tolerance.

## 4. Materials and Methods

### 4.1. Assessment of Soybean ACS Genes

Whole-genome data were obtained from Phytozome v12.1 (https://phytozome.jgi.doe.gov/pz/portal.html (accessed on 28 April 2024)) and ACS protein sequences from *Triticum aestivum* L. were then used to identify soybean ACS proteins using BLASTP (e-value cutoff = 1.0) [32]. SMART (https://smart.emblheidelberg.de/ (accessed on 30 April 2024)) and InterProScan (https://www.ebi.ac.uk/interpro/search/sequence-search (accessed on 30 April 2024)) were employed to assess if the *ACS* domain (PF00155) is present in candidate genes. Furthermore, sequences identified as redundant were removed manually [33,34]. The WoLF PSORT (https://www.genscript.com/wolf-psort.html (accessed on 30 April 2024)) software was used to assess protein subcellular locations. The sequence lengths, isoelectric points, and molecular weights of the ACS protein were evaluated via ExPASy (https://web.expasy.org/compute_pi/ (accessed on 30 April 2024)).

### 4.2. Analysis of Evolution, Synteny, and Genetic Structure

The phylogenetic tree was established using the *ACS* gene’s homologous amino acid sequences in *Oryza sativa* (*OsACS*), *Arabidopsis thaliana* (*AtACS*), *Glycine max* (*GmACS)*, *Triticum aestivum L* (*TaACS*), and *Medicago truncatula* (*MtACS*) with the help of MEGA 5.0 [35]. Subsequently, to validate the conserved domains and perform protein sequence alignment, ClustalW was employed [36]. The intron–exon structure of *GmACS* genes was illustrated via Gene Structure Display Server v2.0 (https://gsds.gao-lab.org (accessed on 3 May 2024)). With the help of the Ensembl Plants (http://plants.ensembl.org/species.html (accessed on 3 May 2024)), syntenic blocks between the *ACSs* of the aforementioned plant species were determined [37].

### 4.3. GmACSs Promoter Analyses

To predict the cis-acting elements, the *GmACS*s promoter sequences, 2000 bp sequences upstream of the nucleotide ‘A’, were acquired from the translation initiation site (ATG) [38,39]. The potential cis-acting elements in the promoter sequence were predicted using the PlantCARE (http://bioinformatics.psb.ugent.be/webtools/plantcare/html/ (accessed on 7 May 2024)) [40]. Subsequently, the TBtools (Vers. 2.056) [41] were employed for classification, visualization, and analysis.

### 4.4. GmACS Expression Analyses

To analyze the expression patterns of GmACS in different soybean tissues, samples were collected from various soybean tissues (roots, leaves, pods, seeds, early-seed maturity stage (EM), mid-seed maturity stage (MM), late-seed maturity stage (LM), dry-seed stage (DS), and flowers). cDNA from these samples was used as a template for real-time quantitative RT-PCR amplification, with the expression level of the soybean ACS family genes in roots serving as a control [41]. Additionally, 2nd trifoliolate stage soybean (DN50) seedlings were grown in control (0 mM of any treatment agent), NaHCO_3_ (100 mmol/L), jasmonic acid (JA: 100 μmol L^−1^), mannitol (200 mmol/L), abscisic acid (ABA: 100 μmol L^−1^), PEG6000 (20%), NaCl (150 mmol/L), salicylic acid (SA: 100 μmol L^−1^), and brassinolide (BR: 100 μmol L^−1^) to evaluate the *GmACS*s transcriptional profiles under abiotic stress and hormone treatments. Samples were taken at 0, 1, 3, 6, 12, and 24 h after treatment for real-time quantitative RT-PCR amplification, with the expression levels of the soybean *ACS* family genes at 0 h serving as the control [42].

Total RNA was harvested from the acquired samples via a Quick Total RNA Isolation Kit (HUAYUEYANG, Beijing, China), quantified, and tested for quality. Then, RNA’s structural integrity was evaluated via 1% agarose gel electrophoresis. Subsequently, it was utilized for preparing first-strand cDNA using the Prime Script^®^ RT Master Mix (TaKaRa Dalian, China). The cDNA was then diluted to 500 ng/L, and subjected to qRT-PCR [43] using SYBR Green (TaKaRa) on a Roche LightCycler^®^ 96 q-PCR system. The PCR reaction (20 μL) comprised 10 μL of 2× RT-PCR Mix (containing SYBR Green I), 0.5 μL reverse and forward primers, respectively, and diluted cDNA. The mRNA levels of soybean root under natural settings were utilized for calibrating *GmACS* transcript levels. For each sample, three biological replicates were separately assessed. *GmACTIN4* was employed as a normalization control and the relative gene expression was elucidated via the 2^−∆∆CT^ protocol [44,45]. The statistical significance was determined using Student’s *t*-tests or ANOVA, depending on the experimental design, to compare gene expression levels between different treatment groups and controls. All experiments were performed with at least three biological replicates, and data are presented as mean ± standard deviation (SD). A *p*-value of <0.05 was considered statistically significant. Appendix A lists the primers employed for qRT-PCR.

### 4.5. Assessment of Subcellular Localization

The complete coding regions of the *GmACS* genes were amplified from the roots of the soybean cultivar ‘DN50’. For establishing the fusion proteins, *GmACSs*::GFP, the genes were incorporated into the pBI121 vector with a GFP tag. The *GmACS*s were cloned with the help of gene-specific primers (Appendix A). Subsequently, after 16 h of co-transformation mediated by PEG-calcium, the transient expression of the recombinant *GmACSs*::GFP plasmids was induced in Arabidopsis mesophyll protoplasts [46]. The positive control was an empty vector, whereas Mito-Tracker Red CMXRos (C1049B, Beyotime, Shanghai, China) was employed as an index to evaluate mitochondria [47,48]. The GFP-labeled protein’s subcellular localization was assessed using an LSM710 confocal laser scanning microscope (Zeiss, Oberkochen, Germany).

### 4.6. Enzyme Kinetics

The coding sequence (CDS) of *GmACS15* was cloned into the *BamHI* region of the bacterial expression vector pGEX-4T-3. The pGEX-4T-3-*GmACS15* plasmid was inserted into *Escherichia coli’s* Rosetta strain to generate putative recombinants. After sequencing, the successful clone was propagated for 4 h in LB media (liquid) augmented with IPTG (1 mmol/L) at 37 °C to promote *GmACS15* expression. Then, using the glutathione sepharose high-performance affinity chromatography, the GST-tagged GmACS15 proteins were purified. For Western blotting, anti-GST antibodies were employed. The purified protein’s ACS enzyme activity was evaluated spectrophotometrically using PLP as a coenzyme to catalyze the conversion of SAM to ACC at 450 nm as mentioned previously [28]. Progress curves were generated to acquire the Vmax and KM for SAM or PLP by using different doses of SAM (0–2 mmol/L) or PLP (0–0.5 mmol/L), respectively. GraphPad Prism v8.0.1 was employed to determine the kinetic parameters of *GmACS15* using the Michaelis–Menten equation.

### 4.7. Overexpression of GmACS15 in Soybean Hairy Roots (HR)

To establish an overexpression vector pSOY1-*GmACS15*, the *GmACS15* CDS was cloned into pSOY1 harboring a CaMV35S promoter (primers in Appendix A). Then, the recombination pSOY1-*GmACS15* plasmid from Agrobacterium rhizogene strain K599 was utilized to electroporate soybean hypocotyls. The previously described protocol [49] was followed for hairy root induction and soybean transformation in ‘DN50’ hypocotyls. K599-infected soybean plant’s HRs were used as a control. PCR and enzyme activity assays were carried out on the transgenic HRs, while non-transgenic lines were dissected. To elucidate how transgenic HRs influence various physiological and biochemical variables, >10–20 transgenic HRs were analyzed. Moreover, 5 days after salt stress, the impact of NaCl (0 and 120 mmol/L) on the dry root, dry leaf, fresh root, and fresh leaf weights, as well as maximum root length, and the plant height of soybean’s transgenic HRs was assessed.

### 4.8. Staining and Biochemical Physiological Parameters Analyses

The plant height and root length of the transgenic HR samples after 5 d of NaCl treatments (0 or 120 mmol L^−1^) were assessed. The seedling roots were isolated from the upper part of the roots, placed at 105 °C for 15 min, and dried at 80 °C to constant weight. Then, the dry weight of each seedling’s above and underground parts was evaluated, respectively.

For staining, different wild-type and overexpressed leaves from the 0 and 2 h stress treatments were placed in 2 mL centrifuge tubes comprising NBT, DAB (1.5 mL), and Evan’s blue staining solution overnight at ambient temperature. After staining, samples were subjected to decolorization with 75% ethanol +5% glycerin in a boiling water bath [50]. For PI staining, root tips of 12 h stress treatment wild-type and overexpressed plants were reacted with PI staining solution (20 μg/mL) for 30 min, rinsed thrice with distilled water and then observed and imaged via a laser confocal microscope [51]. In addition, the root tips of CHR and OHR plants were treated under two conditions: either no treatment or 12 hours of stress treatment. Subsequently, all samples were stained with the CoroNa™ Green Sodium Indicator staining solution for 30 minutes [52], rinsed thrice with distilled water, and then, observed and imaged via a confocal laser microscope.

The superoxide (O_2_^−^) and hydrogen peroxide (H_2_O_2_) levels were assessed via Spectrophotometry [53,54]. The lipid peroxide (malondialdehyde, MDA) content was elucidated using the TBARs (thiobarbituric acid reaction) method [55]. The soybean extracts (0.5 g) were prepared with buffer (15 mL) comprising PVP (1%: *w*/*v*), K_2_HPO_4_-KH_2_PO_4_ (50 mmol/L; pH 7.0), EDTA (1.5 mmol/L), and ASA (0.5 mmol/L) to assess their capability of scavenging reactive oxygen species (ROS). The activities of antioxidant enzymes including peroxidase (POD), superoxide dismutase (SOD), and catalase (CAT) were assessed in the resultant supernatant [56,57].

## 5. Conclusions

In summary, this research study identified 22 *ACS* genes in the soybean genome. These *ACS* genes were categorized into four phylogenetic subfamilies per phylogenetic analysis and subcellular localization. Furthermore, it was observed that during salt stress, the potential functional variations across the *GmACS* isoforms were mirrored in their differential expression. *GmACS15’s* transcriptional response to salinity revealed that it might be the key gene associated with the salt stress response of soybean plants. Therefore, it was inferred that *GmACS15* can decrease salt-induced oxidative damage by modulating the ROS-scavenging system, cellular redox homeostasis, and maintaining intracellular Na^+^/K^+^ homeostasis. These findings provide a theoretical basis for strategies to increase soybean’s genetic salt tolerance. In addition, the function of other *ACS* genes except *ACS15* in soybean stress response remains to be further studied. Therefore, it is necessary to selectively silence the *GmACS* gene and study its physiological and biochemical characteristics.

## Figures and Tables

**Figure 1 ijms-26-02526-f001:**
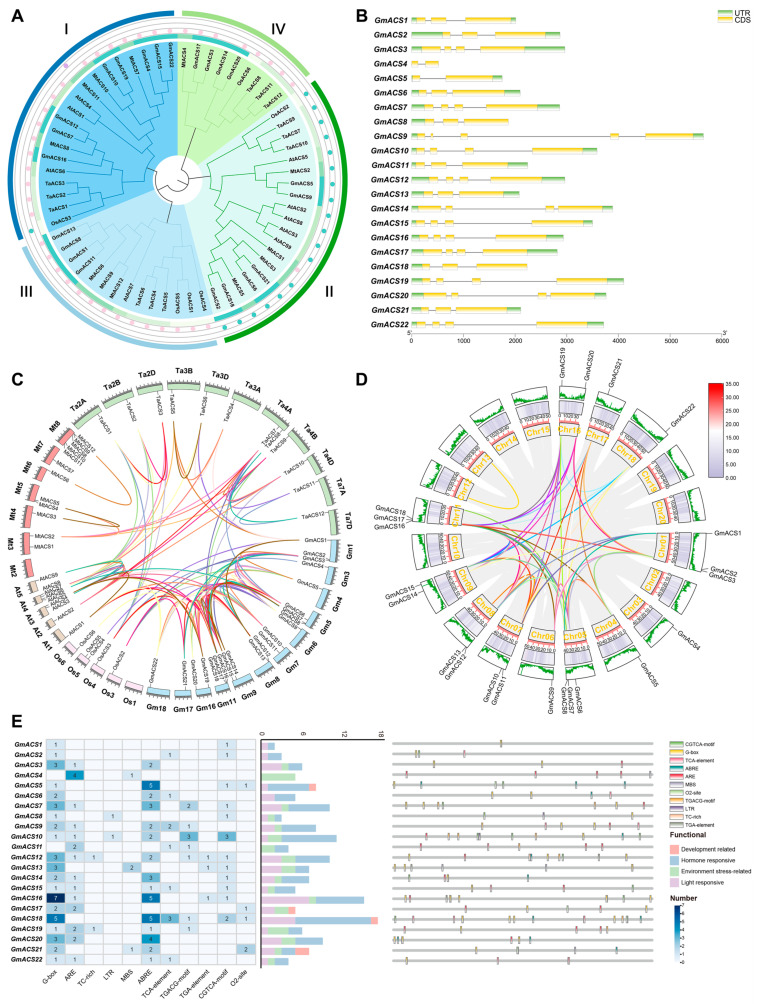
Phylogenetic and gene structure analysis of *GmACS*s. (**A**) Evolutionary analysis of ACS proteins from *Arabidopsis thaliana*, *Glycine max*, *Oryza sativa*, *Medicago truncatula*, and *Triticum aestivum*. (**B**) Gene structure of *ACSs*. The yellow box represents the gene CDS. The green box represents the UTR sequence. (**C**) Syntenic analysis of *Arabidopsis thaliana*, *Glycine max*, *Oryza sativa*, *Medicago truncatula*, and *Triticum aestivum ACS* genes (the length of *Triticum aestivum* chromosome is one-tenth of the true length). (**D**) Syntenic analysis of soybean *GmACS* genes. Chromosomes are shown as circles. The curves of different colors represent syntenic areas of *ACS* genes. The size of exons and introns can be estimated from the bottom scale. (**E**) The classification and statistical analysis of cis-acting elements are presented. Eleven types of cis-acting elements are included, such as light-responsive elements (purple), hormone stress elements (blue), environment stress-related elements (green), and plant growth and development elements (pink). In the left grid, each element’s count is represented numerically, with a color gradient from white to blue indicating an increase in element count. The bar chart on the right counts the total number of different categories of cis-acting elements in each *GmACS* gene. The distribution of cis-acting elements in the promoter region of *GmACS* genes (−2000 bp) is represented by different colors.

**Figure 2 ijms-26-02526-f002:**
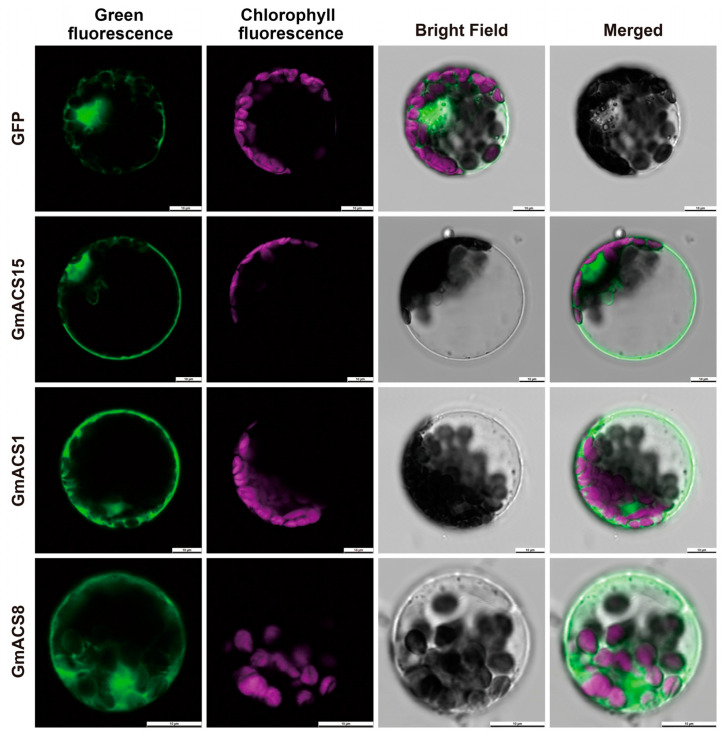
An analysis of the subcellular localization of *GmACS*s. In *Arabidopsis* mesophyll protoplasts, 3 GmACS-GFP fusion proteins were transiently expressed. Confocal micrographs indicate the subcellular location of the GmACS1, -8, and -15 proteins. Bars = 10 μm.

**Figure 3 ijms-26-02526-f003:**
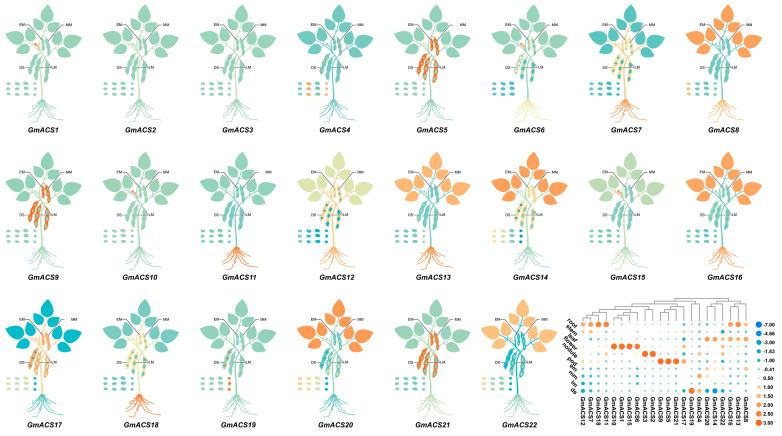
*GmACS*s transcriptional expression in various tissues. The heatmap was generated using the TBtools based on log2 expression values. Blue = low transcription levels; orange = high transcription levels.

**Figure 4 ijms-26-02526-f004:**
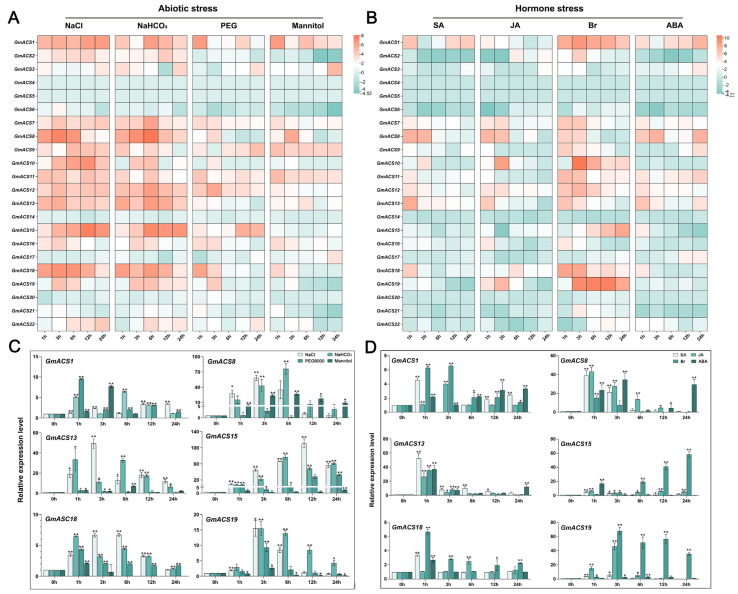
*GmACS*s expression profile under abiotic stress and hormone stimulations and alterations in candidate genes under stress. (**A**) The expression patterns of *GmACS* were assessed in soybean plant roots treated with NaCl (150 mmol L^−1^), PEG (20%), mannitol (200 mmol L^−1^), NaHCO_3_ (100 mmol L^−1^), or water for 0, 1, 3, 6, 12, and 24 h, respectively. (**B**) The expression patterns of *GmACS* were assessed in soybean plant roots treated with SA, ABA, BR, or JA (100 μmol L^−1^) or water as a control at 0, 1, 3, 6, 12, and 24 h post stimulation. The expression acquired in the absence of stress was employed as a reference control. (**C**) Candidate genes’ expression patterns during abiotic stress and (**D**) hormone stimulation conditions. Asterisks above the bars indicate significant differences, as shown by *t*-tests (** *p* < 0.01; * *p* < 0.05).

**Figure 5 ijms-26-02526-f005:**
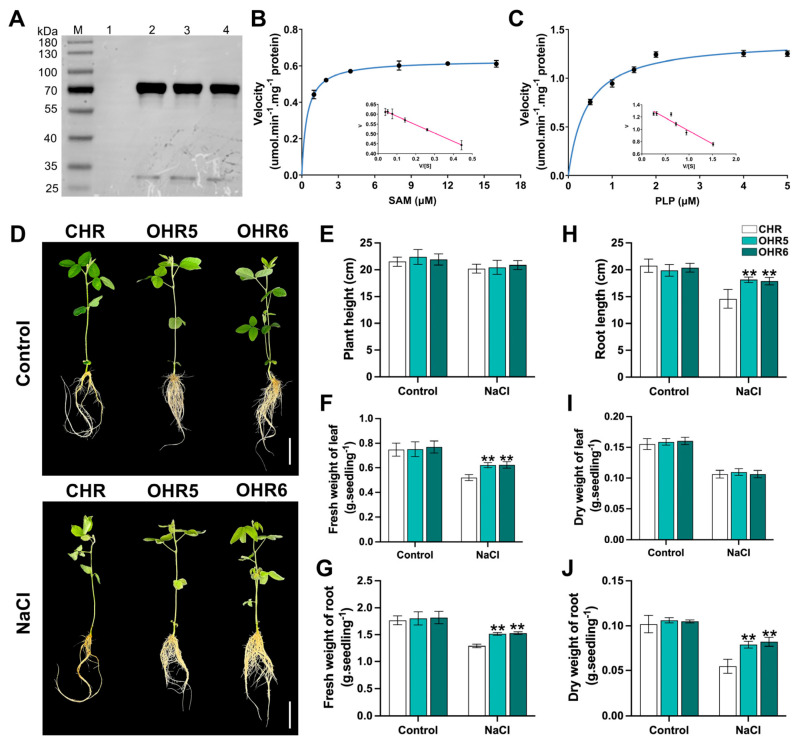
*GmACS15* overexpression increases the tolerance of transgenic soybean HRs to salt. (**A**) Western blotting showing the empty vector pGEX-4T-3 in *E. coli* (Lane 1) and the purified GmACS15 fusion protein (Lanes 2, 3, 4). Lane M, protein markers. Eadie–Hofstee plots were used to determine GmACS15 kinetics using the substrates (**B**) PLP and (**C**) SAM. (**D**) Performance, (**E**) plant height, (**F**) leaf fresh weights, (**I**) leaf dry weights, (**H**) root length, (**G**) root fresh weight, and (**J**) root dry weight of control soybean HRs (CHR) and *GmACS15* overexpressing soybean HRs (OHR) plants subjected to 120 mmol/L NaCl over 5 d. Bars = 5 cm. Asterisks indicate significant differences (** *p* < 0.01) from CHR plants, assessed by Student’s *t*-test.

**Figure 6 ijms-26-02526-f006:**
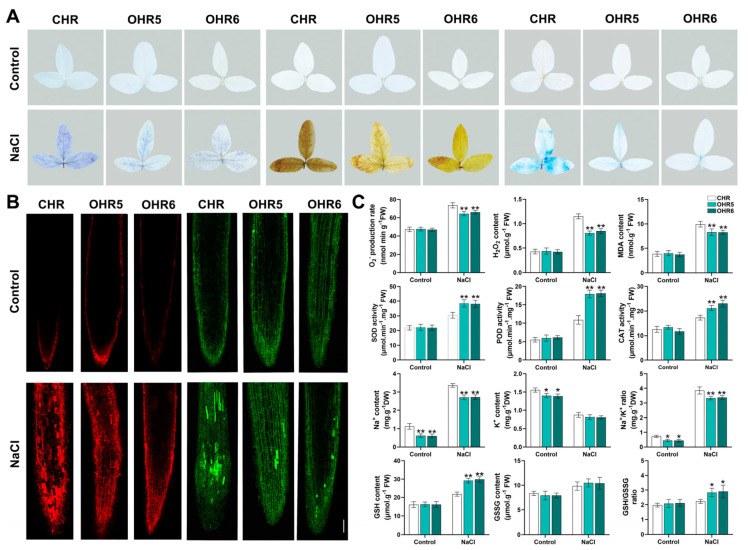
*GmACS15* elevates the antioxidant activity of soybeans during salt or osmotic stress. (**A**) Representative images of NBT, DAB, and Evan’s blue staining of *GmACS15*-OHR and CHR plant leaves subjected to 0 or 120 mmol/L NaCl for 24 h. (**B**) PI and CoroNa™ Green staining of GmACS15-OHR and CHR plant roots subjected to NaCl (0 or 120 mmol/L) for 12 h. Bars = 100 μm. (**C**) O_2_^−^, H_2_O_2,_ MDA, Na^+^, and K^+^ contents and activities of CAT, GSH, POD, SOD, and GSSG in the root of CHR and OHR plants treated as above. The asterisks depict significant differences (** *p* < 0.01; * *p* < 0.05) from CHR plants, acquired by Student’s *t*-test.

**Figure 7 ijms-26-02526-f007:**
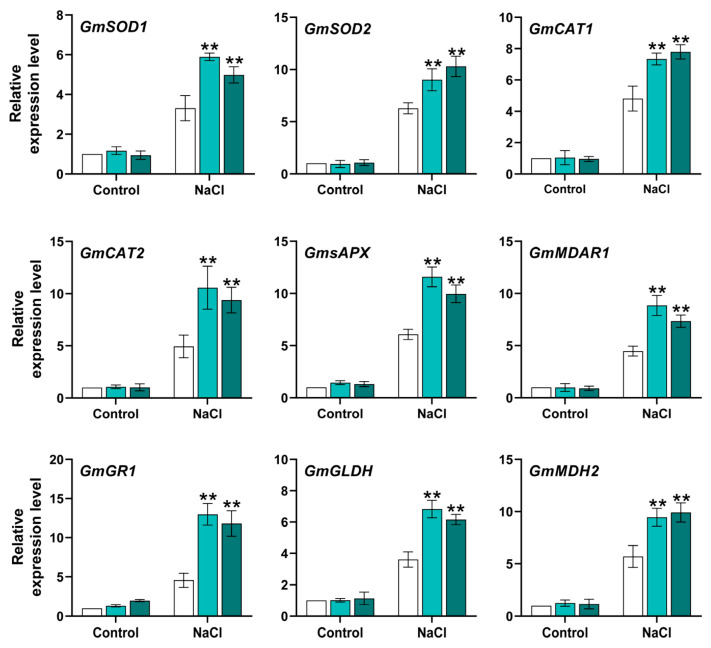
Transcripts of core genes associated with ROS homeostasis and cellular redox in *GmACS15* transgenic HRs. The mRNA level of redox homeostasis genes (GR1, GLDH, MDAR1, and sAPX) and antioxidant system-related genes (SOD1, CAT1, CAT2, PER22, and SOD2) in CHR and OHR subjected to NaCl (0 or 120 mmol/L for 12 h). Values are plotted as means ± SEs (*n* = 3). The asterisks depict Student’s *t*-test significant differences (** *p* < 0.01) from CHR.

## Data Availability

Data may be made available upon reasonable written request to the corresponding author.

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
