# Peer review of "Genome-Wide Characterization of Soybean 1-Aminocyclopropane-1-carboxylic Acid Synthase Genes Demonstrates the Importance of GmACS15 in the Salt Stress Responses"

_ijms, 2025, doi:10.3390/ijms26062526_

Round 1

Reviewer 1 Report

Comments and Suggestions for Authors

As a non-expert in this type of analysis, I found the work very easy to read and described the analyses and results with certainty.
but I have some questions.
Why was soybean chosen as the crop for this study?
I would like to know the origin of the materials used and the experimental plan implemented to obtain salt stress.
Please improve these aspects.

Author Response

非常感谢您花时间审阅此手稿。我们非常感谢您就如何解决手稿中的某些缺陷提出令人信服的建议。我们仔细研究和处理了您的建议并修改了手稿。我希望您对修改后的手稿感到满意,并期待很快收到您的来信。

评论 1: [为什么选择大豆作为本研究的作物?

回答 1:[感谢您关于选择大豆作为本研究作物的问题。选择大豆 (Glycine max (L.) Merr.) 有几个关键原因:选择大豆作为本研究的作物,因为它作为主要的油籽和蛋白质来源具有重要的经济意义,它对盐胁迫的敏感性,以及土壤盐碱化对农业生产力的全球影响。作为一种对盐敏感的作物,大豆是阐明耐盐性分子机制的理想模型。此外,其丰富的遗传多样性和基因组学的最新进展为鉴定和利用耐盐基因提供了坚实的基础。通过关注大豆,我们的研究旨在为耐盐品种的开发做出贡献,提高受盐碱影响地区的农业生产力,并提供适用于其他作物的见解。]

评论 2: [我想知道所用材料的来源以及为获得盐胁迫而实施的实验计划。

回复 2:[感谢您关于我们研究中所用材料的来源和为获得盐胁迫而实施的实验计划的问题。 以下是详细信息:使用的具体品种为 DN50。大豆种子由实验室工作人员种植和收获,并在我们的温室设施中在受控条件下发芽。为了诱导盐应激,我们遵循了类似研究中使用的成熟方案。引用的参考文献是:大豆苹果酸脱氢酶基因的全基因组表征揭示了 GmMDH2 在盐胁迫反应中的积极作用。并且我们修改了手稿并引用了此参考文献,可以找到此更改 – 4.4 GmACS 表达分析,第 476 行。]

Reviewer 2 Report

Comments and Suggestions for Authors

Review on the manuscript ID ijms-3451899 entitled “Genome-Wide Characterization of Soybean 1-Aminocyclopropane-1-Carboxylic Acid Synthase Genes Demonstrates the Importance of GmACS15 in Salt Stress Responses”. The manuscript presents a comprehensive genome-wide analysis of ACS genes in soybean, with a strong focus on GmACS15 and its role in salt stress responses. The research combines bioinformatics, gene expression profiling, and functional validation through transgenic experiments, making it a valuable contribution to the field of plant stress biology. The study is well-structured, and the methodology is appropriate for addressing the research objectives. However, there are some areas that require improvement.
Major comments:
-    In the introduction, the background on ACS genes in other plant species is well-covered, but more background on previous work in soybean would help contextualize the novelty of this study.
-    The discussion on gene expression under stress conditions should provide more in-depth insights into potential regulatory mechanisms.
-    In the methods, write more details on statistical analyses used for gene expression studies should be included.
Minor comments:
Line 34: differences in GmACS15 were notably related to seedling salt 
Line 38:  enhancing salt tolerance in soybean.
Line 42: Soybean, as one of the most important crops globally, plays
Line 141: further classified into four distinct groups
Line 165: these bioinformatics results suggest that

Author Response

Thank you very much for taking the time to review this manuscript. We greatly appreciate your convincing suggestions on how to address certain deficiencies in the manuscript. We have carefully studied and processed your suggestions and revised the manuscript. I hope you will be satisfied with the revised manuscript, and look forward to hearing from you soon.

Comments 1: [In the introduction, the background on ACS genes in other plant species is well-covered, butmore backaround on previous work in soyybean would help contextualize the novelty of this study.]

Response 1: [Thank you for your valuable feedback. We appreciate your suggestion to provide more background on previous work related to ACS genes in soybean to better contextualize the novelty of our study. In response, we have revised the introduction to include additional information on the existing research on ACS genes in soybean. We believe that these revisions effectively contextualize our study within the existing body of research on ACS genes in soybean and highlight its novelty and significance. This change can be found – Introduction, paragraph 5.]

Comments 2: [The discussion on gene expression under stress conditions should provide more in-depth insights into potential regulatory mechanisms.]

Response 2: [Thank you for your suggestion to provide more in-depth insights into the potential regulatory mechanisms underlying gene expression under stress conditions. We have revised the discussion section to incorporate recent findings and theoretical frameworks that elucidate the complex regulatory mechanisms involved in stress responses. We believe that these revisions provide a more comprehensive discussion of the potential regulatory mechanisms underlying gene expression under stress conditions and highlight the complexity of the stress response in plants.This change can be found – Discussion, paragraph 3, lines 392-400.]

Comments 3: [In the methods, write more details on statistical analyses used for gene expression studies should be included.]

Response 3: [Thank you for your suggestion to provide more details on the statistical analyses used for gene expression studies in our manuscript. We have revised the methods section to include a comprehensive description  in our real-time quantitative RT-PCR (qRT-PCR) experiments. This change can be found –4.4GmACS expression analyses, paragraph 2, lines 488-492. We hope these revisions meet your expectations and provide a more comprehensive context for our study.]

Comments 4: [Line 34: differences in GmACS15 were notably related to seedling salt ]

Response 4: [We appreciate your suggestion to amend the sentence “differentiations in GmACS15 were notably related to seedling’s salt tolerance” to “differences in GmACS15 were notably related to seedling salt”. However, upon re-evaluating the content of this sentence, we realized that it does not directly contribute to the core focus of our study.  Therefore, we have decided to delete this sentence to maintain the clarity and relevance of our manuscript.]

Comments 5: [Line 38: enhancing salt tolerance in soybean. ]

Response 5: [Thank you for your careful review and feedback on our manuscript. We have made the necessary corrections to address the issues raised, including the typographical error in the sentence regarding “enhancing salt tolerance in soybean.” The extra period has been removed, and we have also conducted a thorough check of the manuscript to ensure no other similar errors remain. This change can be found – Abstract, line 35.]

Comments 6: [Line 42: Soybean, as one of the most important crops globally, plays ]

Response 6: [We appreciate your suggestion to add “important” to emphasize the importance of soybeans. We have revised the sentence. The revised sentence now reads:“Soybean, as one of the most important crops globally,”. This change can be found – Introduction, line 39.]

Comments 7: [Line 141: further classified into four distinct groups ]

Response 7: [We appreciate your suggestion to use the numeric form "4" instead of the word "four" in our text. We understand that this change aligns with the preferred style and consistency of the manuscript. We have revised the sentence to use the numeric form "4" instead of the word "four.". This change can be found – 2.3  Regulatory elements in the GmACS promoters, line 147.]

Comments 8: [Line 165: these bioinformatics results suggest that ]

Response 8: [Thank you for your constructive feedback on our manuscript. We have carefully considered your suggestion regarding the use of "involved" instead of "crucial" in our description of the bioinformatics results. We agree that "involved" is a more appropriate term to describe the role of GmACSs based on our current findings. We have revised the sentence. The revised sentence now reads:“Overall, these bioinformatics results suggested that GmACSs are involved in regulating plant development and stress responses.”. This change can be found – 2.3  Regulatory elements in the GmACS promoters, line 161.]

Reviewer 3 Report

Comments and Suggestions for Authors

This contribution is an good example of the use of bioinformatics to confirm already described physiological evidence of the role of ethylene in abiotic stress response. Few changes are suggested in the file. 

Comments on the Quality of English Language

Few changes are needed, the writing is good. 

Author Response

Thank you very much for taking the time to review this manuscript. We greatly appreciate your convincing suggestions on how to address certain deficiencies in the manuscript. We have carefully studied and processed your suggestions and revised the manuscript. I hope you will be satisfied with the revised manuscript, and look forward to hearing from you soon.

Comments 1: [Cite the current soybean consumption, including the reference.]

Response 1: [Thank you for your suggestion to include current data on soybean consumption. We have revised the manuscript to incorporate the latest information on global soybean consumption patterns and their implications. In the revised manuscript, we have incorporated the following information:“According to recent data, soybeans are primarily consumed in three main ways: direct human consumption (20%), animal feed (76%), and industrial uses (4%).”. The cited reference is: Soybean prices and sustainability. This change can be found – Introduction, line40-42.]

Comments 2: [Cite the estimate of adverse effect in soybean production, including a reference.]

Response 2: [Thank you very much for your valuable suggestion to include a citation for the estimate of adverse effects on soybean production. We have revised the manuscript accordingly and now provide specific data and references to support the discussion on the impact of biotic and abiotic stresses on soybean yield and quality. In the revised manuscript, we have incorporated the following information:“It has been estimated that drought stress alone can reduce soybean yields by up to 50% in affected regions.”. The cited reference is: Climate Change: Projections and Its Possible Impact on Soybean. This change can be found – Introduction, line43-45.]

Reviewer 4 Report

Comments and Suggestions for Authors

Introduction: The work is important, but the introduction is weak and does not clearly and in-depth report the topic to be addressed. It is necessary to insert the objective of the study

Lines42-46: It would be interesting to focus on the stress analyzed in this work

Line 62:  Check this information. We check works with this theme: Tucker ML, Xue P, Yang R. 1-aminocyclopropane-1-carboxylic acid (ACC) concentration and ACC synthase expression in soybean roots, root tips, and soybean cyst nematode (Heterodera glycines)-infected roots. J Exp Bot. 2010;61(2):463–72; 

Lines 83-84: Verify this information.

Lines 86-91: The results of this research should not be presented in the introduction.

Line 419: The methodology does not describe the treatments that induced stress in the plants.

Line 102: Table 1: Unidentified

Lines: 192-194: And how were these plants obtained?

Lines 197-199: methodology not identified

Lines 212-213: this is discussion and not result.

Lines254-255: indicate in the methodology

Line342: An in-depth review is needed to confirm this statement

Author Response

Thank you very much for taking the time to review this manuscript. We greatly appreciate your convincing suggestions on how to address certain deficiencies in the manuscript. We have carefully studied and processed your suggestions and revised the manuscript. I hope you will be satisfied with the revised manuscript, and look forward to hearing from you soon.

Comments 1: [ The work is important, but the introduction is weak and does not clearly and in-depth report the topic to be addressed. It is necessary to insert the objective of the study]

Response 1: [Thank you for your constructive feedback on our manuscript. We appreciate your comments regarding the need to strengthen the introduction and clearly state the objective of our study. We have carefully revised the introduction to address these concerns and provide a more in-depth background and clear objectives. Therefore we have revised the text to better articulate the aims and significance of our study. The revised passage now reads: our study aims through genome - wide analysis and research on expression patterns under multiple stress conditions, the structural characteristics, evolutionary relationships, and functional divergence of the soybean ACS gene family under various stress conditions were systematically analyzed. This not only fills the gap in the systematic study of the soybean ACS gene family but also provides new perspectives and theoretical bases for a deeper understanding of their roles in plant adaptation to complex environmental changes. This change can be found – Introduction, lines 90-97.]

Comments 2: [Lines42-46: It would be interesting to focus on the stress analyzed in this work]

Response 2: [Thank you very much for your insightful suggestion to focus on the stress analysis in this work. We fully agree that a detailed investigation of stress factors and their impact on soybean growth is of great significance, especially in the context of global climate change. Therefore, we have revised the manuscript to place greater emphasis on the analysis of biotic and abiotic stresses and their effects on soybean. This change can be found – Introduction, lines 39-50.]

Comments 3: [Line 62:  Check this information. We check works with this theme: Tucker ML, Xue P, Yang R. 1-aminocyclopropane-1-carboxylic acid (ACC) concentration and ACC synthase expression in soybean roots, root tips, and soybean cyst nematode (Heterodera glycines)-infected roots. J Exp Bot. 2010;61(2):463–72; ]

Response 3: [Thank you for pointing this out.We have thoroughly reviewed the information related to Tucker et al. (2010) and have ensured that our manuscript accurately reflects their findings while clearly delineating the novel contributions of our research. This change can be found – Introduction, line 88.]

Comments 4: [Lines 83-84: Verify this information.]

Response 4: [Thank you for your constructive feedback on our manuscript. We have carefully reviewed and revised the relevant information. We have updated the relevant references to ensure their accuracy and relevance. The revised passage now reads: Consequently, numerous ACS genes have emerged as a focal point of research. Nevertheless, the functional characterization of ACS genes in soybean in response to abiotic stresses, such as salinity, remains equivocal. This change can be found – Introduction, lines 64-66. ]

Comments 5: [Lines 86-91: The results of this research should not be presented in the introduction.]

Response 5: [We apologize for any confusion caused by presenting the results in the introduction section.   We understand that the introduction should focus on setting the stage for the research, including background information, the problem statement, and the objectives of the study, rather than presenting the results. We have thoroughly reviewed the introduction section and removed any premature mention of the results.  ]

Comments 6: [Line 419: The methodology does not describe the treatments that induced stress in the plants.]

Response 6: [Thank you for your feedback regarding the description of stress treatments in our methodology. We describe in detail the treatment of induced plant stress in 4.4. GmACS expression analyses. In our study, we employed several stress treatments to investigate the regulation and functional roles of GmACS genes in soybean under different environmental conditions. The treatments included: NaHCO3 (100 mmol/L), jasmonic acid (JA: 100 μmol L-1), mannitol (200 mmol/L), abscisic acid (ABA: 100 μmol L-1), PEG6000 (20%), NaCl (150 mmol/L), salicylic acid (SA: 100 μmol L-1), and brassinolide (BR: 100 μmol L-1) to evaluate the GmACSs transcriptional profiles under abiotic stress and hormone treatments. These treatments were selected based on their relevance to common abiotic stress conditions that soybean plants encounter in the field and their ability to induce significant responses in gene expression and physiological traits. ]

Comments 7: [Line 102: Table 1: Unidentified]

Response 7: [Thank you for pointing out the issue regarding Table 1. We apologize for the oversight and have made changes.  The word “Table 1” has been replaced by “Table S1” as you advised in the newly submitted manuscript. This change can be found – Results, line 107.]

Comments 8: [Lines: 192-194: And how were these plants obtained?]

Response 8: [Thank you for your question regarding the source of the plants used in our study. The plants were obtained through standard laboratory and greenhouse protocols. Specifically, soybean seeds served as the primary material and were grown and harvested by laboratory workers for genetic transformation. These seeds were then germinated under controlled conditions in our greenhouse facility. The seedlings were then grown to maturity under optimized growth conditions to ensure uniformity and health before being subjected to the experimental treatments.]

Comments 9: [Lines 197-199: methodology not identified]

Response 9: [Thank you for your valuable comments and suggestions regarding our manuscript. We have revised the Methods section to provide a comprehensive and detailed description of our research methods. We hope that these revisions have addressed the concerns regarding the methodology. We believe that these improvements will enhance the clarity and transparency of our research approach. This change can be found – 4.4. GmACS expression analyses, lines 464-476.]

Comments 10: [Lines 212-213: this is discussion and not result.]

Response 10: [Thank you for your insightful comments regarding the content on Lines 212-213.  We fully agree that the content in these lines was more aligned with discussion rather than results. Therefore, we removed it from the results section for this issue.]

Comments 11: [Lines254-255: indicate in the methodology]

Response 11: [Thank you for your comment regarding Lines 254-255.  We appreciate your attention to the clarity and completeness of our manuscript. We would like to confirm that the details related to the content on Lines 254-255 have already been thoroughly described in the Methodology section.  Specifically, the relevant information can be found in the subsection titled “[4.7. Overexpression of GmACS15 in soybean hairy roots (HR)]” on Lines [520-524] of the Materials and Methods. ]

Comments 12: [Line342: An in-depth review is needed to confirm this statement]

Response 12: [Thank you for your suggestion regarding the statement on the ACS gene family in soybeans.  We have conducted an in-depth review to confirm the accuracy of this statement and provide a more comprehensive context.
 “Recent investigations have delved into certain facets of the ACS gene family in soybeans. Nevertheless, a comprehensive and in - depth comprehension of the entirety of the ACS gene family in soybeans has yet to be achieved, thus necessitating further exploration.”. This change can be found – Discussion, lines 346-348. ]